# Early Complications of Radioisotope Therapy with Lutetium-177 and Yttrium-90 in Patients with Neuroendocrine Neoplasms—A Preliminary Study

**DOI:** 10.3390/jcm11040919

**Published:** 2022-02-10

**Authors:** Barbara Bober, Marek Saracyn, Kornelia Zaręba, Arkadiusz Lubas, Paweł Mazurkiewicz, Ewelina Wilińska, Grzegorz Kamiński

**Affiliations:** 1Department of Endocrinology and Isotope Therapy, Military Institute of Medicine, 04-141 Warsaw, Poland; barbara.bober@tlen.pl (B.B.); msaracyn@wim.mil.pl (M.S.); gkaminski@wim.mil.pl (G.K.); 2First Department of Obstetrics and Gynecology, Centre of Postgraduate Medical Education, 01-813 Warsaw, Poland; 3Department of Internal Diseases, Nephrology and Dialysis, Military Institute of Medicine, 04-141 Warsaw, Poland; alubas@wim.mil.pl; 4Laboratory of Ethology, Nencki Institute of Experimental Biology PAS, 02-093 Warsaw, Poland; pawel.j.mazurkiewicz@gmail.com; 5Department of Medical Diagnostics, Military Institute of Medicine, 04-141 Warsaw, Poland; ewilinska@wim.mil.pl

**Keywords:** neuroendocrine neoplasm, treatment of neuroendocrine neoplasm, PRRT

## Abstract

Neuroendocrine neoplasms (NENs) constitute a heterogenous group of tumors originating from neuroendocrine cells scattered throughout the body. Peptide Receptor Radionuclide Therapy (PRRT) is a treatment of choice of unresectable metastasized progressive and well-differentiated NENs. The aim of the study was to assess early bone marrow and kidney injury after administration of Lutetium-177 or Lutetium-177 combined with Yttrium-90. Thirty-one patients received treatment with [^177^Lu]Lu-DOTATATE with the activity of 7.4 GBq. Eleven patients received tandem treatment with [^90^Y]Y-DOTATATE with the activity of 1.85 GBq + [^177^Lu]Lu-DOTATATE with the activity of 1.85 GBq. After PRRT a significant decrease in leukocyte, neutrophil, and lymphocyte counts was noted. Tandem treatment demonstrated a more marked decrease in white blood cell count compared to Lutetium-177 therapy only. Conversely, no significant influence on glomerular filtration was found in this assessment. However, PRRT triggered acute renal tubule dysfunction, regardless of the treatment type. Regarding the acute complications, PRRT appeared to be a safe modality in the treatment of patients with NEN.

## 1. Introduction

Neuroendocrine neoplasms (NENs), formerly called neuroendocrine tumors (NET), are a heterogenous group of neoplasms that originate from neuroendocrine cells scattered throughout the body, forming the diffuse endocrine system (DES). They are mostly located in the gastrointestinal and respiratory systems, which are, therefore, their most common locations [1]. Slowly developing tumors constitute the majority of cases, hence they are perceived as rare [2]. However, contemporary imaging techniques contributed to a surge in their detectability over the past few decades [3,4,5,6]. The number of cases of diagnosed neuroendocrine neoplasms has increased over six-fold in the United States since 1973 [7]. The small intestine, particularly the ileum, is the most common location of NEN in the human body [8]. The only method to cure a patient still involves the endoscopic or surgical resection of the tumor [9,10,11]. However, the majority of lesions are detected at high stage, when the radical surgery is impossible to perform, so alternative options are then implemented. The most common methods used in case of advanced lesions include somatostatin analogue therapy, chemotherapy, and targeted therapy [10,12,13,14,15,16,17,18]. 

Peptide Receptor Radionuclide Therapy (PRRT) is a method based on the use of peptides combined with radionuclides emitting beta radiation [19]. In September 2017 (Europe) and in January 2018 (the United States), [^177^Lu]Lu-DOTATATE (Lutathera) was officially accepted for treatment on the basis of the results of the first multicenter randomized phase 3 study (NETTER-1) [20]. PRRT is used in patients with well- and moderately-differentiated unresectable or metastatic NENs with the histological grades G1 and G2, and in some cases G3 as well [21]. Currently, implementation of the tandem therapy is also increasing. De Jong et al. demonstrated that treatment combining 50% of [^90^Y]Y-DOTATATE and 50% of [^177^Lu]Lu-DOTATATE tripled the survival time in rats. The effect was probably due to the synergistic mechanism acting both on minor and major metastatic lesions [22]. Research conducted in people revealed that tandem treatment prolonged the mean survival time compared to treatment based solely on [^90^Y]Y-DOTATATE [23]. The current PRRT treatment regimen involves the administration of 4 cycles of a radioisotope/s at 8- to 12-week intervals. Dose fractionation allows more effective tumor irradiation without exceeding the dose limit of 23 Gy for kidneys. The risk of kidney injury is a factor that limits the use of higher radioisotope activities [19]. The intervals between subsequent doses result from the time that is required for limiting possible myelotoxicity and allowing bone marrow regeneration [19]. 

The most common adverse effects include kidney injury and myelosuppression, but they are relatively rare. Regarding the high survival rate of patients, myelosuppression constitutes a significant adverse event and is more common in individuals in whom cytopenia had been confirmed prior to treatment initiation [24,25,26]. Acute hematologic complications (grade 3 or 4 according to the WHO) occur in fewer than 13% of patients receiving Yttrium-90 and in 10% of those receiving Lutetium-177 [27]. Acute kidney complications mainly depend on the activity of the radiopharmaceutical agents and concomitant diseases [27,28,29,30,31]. Kidney irradiation is associated with radiopeptide reabsorption in the proximal convoluted tubules with their subsequent concentration in the renal interstitium [32,33]. The risk of renal complications is reduced by the administration of amino acids, but it is not fully eradicated [34,35,36,37,38]. According to some authors, the use of PRRT decreased the creatinine clearance by 7.3%/year for Yttrium-90 and by 3.8%/year for Lutetium-177 [39]. However, severe renal injury was an infrequent finding [37]. Long-term research demonstrated that patients with risk factors, such as arterial hypertension, diabetes, or a history of chemotherapy, were more susceptible to the deterioration of glomerular filtration [40]. A NETTER-1 study revealed that myelosuppression occurred in fewer than 10% of those patients [20]. 

The largest group of NEN patients undergoing PRRT in Poland and Central and Eastern Europe is followed in the Department of Endocrinology and Isotope Therapy of the Military Institute of Medicine in Warsaw. Due to the paucity of research and equivocal results, we decided to assess the acute complications in this group of patients. Given this, the aim of the study was to evaluate acute complications (not yet widely assessed in the literature) after radioisotope treatment with Lutetium-177 or Lutetium-177 combined with Yttrium-90.

## 2. Materials and Methods

### 2.1. Materials

The presented paper is a preliminary study to assess the complications of radioisotope treatment of NEN patients. The study group included 42 patients receiving PRRT treatment for NEN between November 2017 and June 2019 in the Department of Endocrinology and Isotope Therapy of the Military Institute of Medicine in Warsaw. We obtained informed consent in writing from all the patients qualified for PRRT. 

The study was approved by the Local Bioethical Committee of the Military Institute of Medicine (52/WIM/2017). All the procedures conducted during the study were compliant with the Declaration of Helsinki of 1964 and its subsequent amendments.

The following inclusion criteria were determined:(a).well- and moderately-differentiated unresectable metastatic progressive neuroendocrine neoplasm (defined as Ki-67 < 20%, progression according to the RECIST 1.1 (Response Evaluation Criteria In Solid Tumours) criteria, over the previous 12 months);(b).good expression of somatostatin receptors in qualifying somatostatin receptor scintigraphy (SRS) (SPECT/CT) (radiotracer uptake in the majority of the lesions higher than in normal liver (Krenning scale ≥ 3)) or in Gallium-68-PET/CT (SUV_max_ in the majority of the lesions higher than SUV_max_ in normal liver);(c).exhausting all the possibilities of surgical treatment;(d).chronic treatment with long-acting somatostatin analogues. 

The exclusion criteria were as follows: (a).no consent to the treatment;(b).pregnancy (a negative pregnancy test was required), lactation;(c).physical fitness assessment (PS, performance status) of the patient based on the WHO/ECOG classification: PS status 3 or 4, or based on the Karnofsky classification (<60); (d).no uptake of the radiotracer in somatostatin receptor imaging (SRI): SRS SPECT/CT or Ga-68-PET/CT;(e).bone marrow failure defined as: hemoglobin below 8 g/dL, platelets below 80 × 10^3^/µL, leucocytes below 2 × 10^3^/µL, lymphocytes below 0.5 × 10^3^/µL, or neutrophils below 1 × 10^3^/µL; (f).creatinine clearance <30 mL/min, blood urea nitrogen (BUN) over 45 mg/dL, and creatinine over 1.8 mg/dL;(g).three-fold increase in total bilirubin concentration;(h).systemic infections;(i).glomerulonephritis;(j).interstitial nephritis;(k).obstructive nephropathy;(l).urinary tract infection.

Thirty-one patients were qualified for treatment with [^177^Lu]Lu-DOTATATE with the activity of 7.4 GBq (200 mCi). In those patients, there were no lesions over 50 mm. They were also characterized by the homogeneous distribution of somatostatin receptors in SRI and/or reduced glomerular filtration (eGFR < 60 mL/min/1.73 m^2^). Eleven patients were qualified for tandem treatment with [^90^Y]Y-DOTATATE with the activity of 1.85 GBq + [^177^Lu]Lu-DOTATATE with the activity of 1.85 GBq. In those patients, at least one lesion was over 50 mm, the distribution of somatostatin receptors was heterogeneous in SRI, and/or GFR was ≥ 60 mL/min/1.73 m^2^.

### 2.2. Methods

The patients were administered intravenous infusions of [^177^Lu]Lu-DOTATATE with the activity of 7.4 GBq or tandem treatment with [^90^Y]Y-DOTATATE + [^177^Lu]Lu-DOTATATE with the activity of 1.85GBq + 1.85 GBq respectively (ItraPol and LutaPol, manufactured by the National Centre for Nuclear Research, Radioisotope Centre POLATOM, Otwock, Poland) (a solution containing radiopharmaceuticals in 100 mL of 0.9% NaCl, a 20-min infusion). The assessment of the morphological and biochemical parameters was performed one day before and two days after radioisotope administration. The patients also received the infusions of 10% amino acid solution (Nephrotect, Fresenius Kabi) (1000 mL on Day 1 and 500 mL on Day 2) and Ringer’s solution (500 mL on Day 1 and 500 ml on Day 2) (Table 1). The treatment with long-acting somatostatin analogues (octreotide—Sandostatin LAR; Novartis and lanreotide autogel-Somatuline; Ipsen) had been discontinued at least four weeks prior to PRRT administration. Previous chemotherapy had to be finished at least three months before.

### 2.3. Morphology and Biochemistry

Morphological and biochemical tests were performed in the Department of Laboratory Diagnostics of the Military Institute of Medicine with the use of Sysmex, XN 1000, 2017 (blood morphology) and Cobas c 501, Roche Diagnostics, 2016 (biochemistry).

Blood tests (Day 1 and 4) included complete blood count with differential, reticulocytes, sodium, potassium, calcium, phosphorus, creatinine, urea, uric acid, ALT, bilirubin, albumins, and KIM-1 (Kidney Injury Molecule-1) concentrations.

### 2.4. KIM-1 in the Serum

The blood was centrifuged (MPW-260R laboratory centrifuge, 10 min, 4000 rpm, temp. +8 °C). The obtained serum was pipetted into an Eppendorf tube (2 mL) and frozen below 80 °C (SimpleFreez™ U80 DH.SWUF0075.E—ultra low temperature (upright) freezer; Daihan Scientific, Jijeong-myeon, Wonju-si, Gang-won-do, Korea) until the test. The measurements were performed with the use of the Human KIM-1 ELISA kit manufactured by Biorbyt. 

### 2.5. Urine

The first morning urine sample was collected for testing. The tests were performed in the Department of Laboratory Diagnostics of the Military Institute of Medicine with the use of Cobas c 501, Roche Diagnostics, 2016. Urine tests (a single sample) (Day 1 and 4) included urinalysis and the sodium, potassium, calcium, phosphorus, creatinine, urea, uric acid, albumins, KIM-1, and IL-18 (Interleukin-18) concentrations.

### 2.6. KIM-1, IL-18 in the Urine

The first morning urine sample was collected for testing. Subsequently, the urine was pipetted into an Eppendorf tube (2 mL) and frozen below 80 °C (SimpleFreez™ U80 DH.SWUF0075.E—ultra low temperature (upright) freezer; Daihan Scientific, Jijeong-myeon, Wonju-si, Gang-won-do, Korea) until the test (approx. 6 months). The measurements were performed with the use of the Human KIM-1 ELISA kit and Human IL18 ELISA kit manufactured by Biorbyt.

### 2.7. Calculated Indices

The estimated glomerular filtration rate (eGFR) was calculated with the mathematical formula recommended by the NKF (the American National Kidney Foundation): the CKD-EPI (Chronic Kidney Disease Epidemiology Collaboration) creatinine equation [41].

### 2.8. Renal Tubule Function Assessment

The assessment of renal tubule function was based on the calculation of the fractional excretion (FE) of: sodium (Lat. Natrium—Na), potassium (Lat. Kalium—K), calcium (Lat. Calcium—Ca), phosphates (PO_4_), uric acid (Lat. Acidum Uricum), and urea (Lat. Urea—U). The formulae used for the calculations are presented in Table 2.

### 2.9. Statistical Analysis

A statistical analysis was performed with the IBM SPSS Statistics package, Version 25.0., Armonk, NY, USA: IBM Corp. (Released 2021). It was used to perform the analyses of basic descriptive statistics with the Shapiro–Wilk test, two-way mixed analysis of variance, and the Mann–Whitney U tests. The criterion for the statistical inference was set at a level of significance of *p* < 0.05.

## 3. Results

### 3.1. Characteristics of the Study Group

Forty-two patients were qualified for the study, including 19 (45.2%) women and 23 (54.8%) men. The mean age of the patients was 58.1 ± 13.1. The mean BMI (Body Mass Index) was 24.9 kg/m^2^, with 50% of patients having normal BMI. The most common concomitant conditions were arterial hypertension (*n* = 18, 42.9% of patients) and diabetes (*n* = 12, 28.6% of patients) (Table 3). Their incidence was significantly higher in the study group than in the general population (42.9% vs. 31.5% in the case of arterial hypertension and 28.6% vs. 9.1% for diabetes). NEN originating from the pancreas (*n* = 15) and NEN originating from the small intestine (*n* = 13) were the most common in the study group. The percentages of patients with G1 and G2 histologic grades were similar (48% vs. 52%, respectively) (Table 3). 

Distant metastases to the liver were found in the marked majority of patients (*n* = 39; 93%). In three remaining cases, the tumors metastasized only to the lymph nodes and bones with the primary NENs originating from the lung, paraganglioma, and one case of undefined origin. Prior to PRRT, six patients had been receiving chemotherapy due to: NEN (*n* = 3), breast cancer (*n* = 1), and colon adenocarcinoma (*n* = 2). Twenty-three (54.8%) cases were functional tumors, including carcinoid syndrome, but there were also rare ones secreting glucagon, growth hormone-releasing hormone (GHRH), and PTH-related protein (PTHrP). The most common location of the primary lesion of functional NEN were the pancreas and small intestine (Table 3). Due to the different aims in this study, we did not reassess the hormonal status of this subgroup of patients because it was established earlier, the patients were treated with long-term somatostatin analogues and most of them were asymptomatic. A majority of patients had undergone surgical resection of the primary tumor and/or metastatic lesions prior to PRRT. In eight cases, the primary tumor was not resected (one patient did not consent to the surgical resection, six patients had unresectable lesions at diagnosis, one person could not undergo surgery because of contraindications for anesthesia). Due to the massive involvement of the liver in metastatic lesions, two patients underwent hemihepatectomy, two underwent thermal ablation, and one underwent the embolization of the lesions in the liver. Prior to the therapy, three patients had undergone simultaneous splenectomy due to the local progression of the primary pancreatic tumor. In this group, there were no patients with MEN-1.

### 3.2. Early Post-PRRT Complications

A statistically significant decrease in leukocyte, neutrophil, and lymphocyte count was observed in the complete blood count after the course of therapy. Changes in individual parameters are presented in Table 4. The sex and BMI of the patients did not influence changes in those parameters. However, it was reported that age was negatively correlated with changes in lymphocyte count. No significant correlations were found between the parameters of complete blood count and the occurrence of diabetes, hypercholesterolemia, or hypertension. The primary origin of NEN and the presence of metastatic lesions in the bones did not influence the complete blood count parameters either. The baseline values of the glomerular filtration rate (GFR) had no effect on the changes in white blood cells. However, the erythrocyte count and hemoglobin were significantly reduced (*p* = 0.01 and *p* = 0.007, respectively) only in the group of patients with baseline GFR < 60 mL/min (Table 5).

The analysis of the renal parameters in the study group after radioisotope administration revealed no significant differences in the assessment of glomerular filtration (Table 5 and Table 6). GFR changes remained unaffected by the age, sex, BMI, history of chronic kidney disease (CKD), diabetes, hypercholesterolemia, hypertension, and the NEN point of origin. 

Furthermore, we observed a statistically significant reduction in the fractional excretion of potassium (FE K%) (*p* < 0.044), an increase in the fractional excretion of urea (FE U%) (*p* < 0.001), and decreased albuminuria (*p* = 0.004), although the albumin-to-creatinine ratio (ACR) remained unchanged (Table 6). In addition, there was a significant reduction in urine IL-18 concentrations (*p* < 0.001) (Table 6). The changes in the above parameters did not depend on age, sex, concomitant diseases, primary NEN point of origin, and type of treatment used. 

All the above changes were associated with the baseline glomerular filtration and were only observed in patients with GFR ≥ 60 mL/min/1.73 m^2^ (Table 5).

With regards to the assessment of liver function, we observed a slight, statistically significant reduction in albumin concentration (*p* < 0.001) and ALT activity (*p* = 0.002), and an increase in bilirubin concentration (*p* = 0.003) (Table 7). However, all these liver parameters remained within a normal range.

An analysis of the complete blood count values and biochemical parameters revealed that the reduction in the leukocyte count was significantly higher (*p* = 0.035) in patients receiving tandem treatment than in those treated with [^177^Lu]Lu-DOTATATE alone (Table 8). Moreover, an association between GFR changes and the type of treatment was observed. The comparison between the group receiving [^90^Y]Y/[^177^Lu]Lu-DOTATATE and the group treated with [^177^Lu]Lu-DOTATATE revealed a slight, but statistically significant, increase in GFR in the former (*p* = 0.018) (Table 8).

### 3.3. Early Post-PRRT Complications According to the CTCAE v. 5.0

The summary of the percentage of patients assigned to the groups of adverse events that occurred after the treatment are presented in Figure 1 and Table 9. The adverse events are categorized according to the Common Terminology Criteria for Adverse Events (CTCAE v. 5.0) of the US National Cancer Institute.

Grade 3 (G3) hematologic complications (decrease of hemoglobin) were reported in only one patient after radioisotope administration. Prior to radioisotope administration, the concentration of hemoglobin in this patient had been consistent with the G2 group. No other grade 3 and 4 hematological and renal toxicity, or any grade liver toxicity, were observed (Table 9).

## 4. Discussion

Most of the reports in the literature on PRRT complications have been devoted almost exclusively to chronic complications. Studies on acute complications are still missing. We focused on such acute complications, probably for the first time in the literature, using very sensitive markers of kidney injury (KIM-1, Il-18) as well as the fractional excretion of various ions as the indicators of renal tubules function. We have thus shown the effect of PRRT on very rare, if ever, demonstrated renal tubular function.

PRRT has been successfully used in NEN patients for two decades. Peptide Receptor Radionuclide Therapy (PRRT) is well-tolerated and safe, but bone marrow and kidneys are organs that limit the use of higher radioisotope activities [42,43,44,45,46]. Our study revealed that the administration of Lutetium-177 or Lutetium-177 and Yttrium-90 led to reduction in the parameters of all hematological lines in the early assessment. It was mainly observed in the number of leukocytes, neutrophils, and lymphocytes. The tandem treatment triggered the more marked suppression of white blood cells, especially leukocytes, compared to treatment with [^177^Lu]Lu-DOTATATE alone. There was no significant reduction in the red blood cell parameters and the platelet count. However, a significant decrease in the number of erythrocytes and the hemoglobin concentration was found in the group of patients with a history of chronic kidney disease. We observed mainly grade 1 and 2 hematologic toxicity according to CTCAE v. 5.0. Grade 3 (G3) hematotoxicity was reported in only one patient after radioisotope administration and it resulted from a shift from G2 to G3. Previous chemotherapy and medical history other than CKD were not related to the changes described above.

Hematotoxicity (bone marrow toxicity) associated with PRRT is mainly due to the irradiation and destruction of the hematopoietic cells. Grade 3 or 4 hematologic toxicity develops in about 5–10% of patients [24,26,29,47,48]. The nadir of radiation activity on the bone marrow usually occurs four–six weeks after radioisotope administration. Currently, the acceptable dose limit for bone marrow is 2 Gy. However, this value was determined in the study of Iodine-131 in thyroid cancer patients [49,50]. Moreover, no unambiguous data are available for [^177^Lu]Lu-DOTATATE to confirm or rule out such a threshold. Immunodeficiency is a serious adverse effect of radiotherapy, especially during treatment with high radioisotope activities. It results from radiation-related cytotoxicity on bone marrow cells and the immunocompetent cells in peripheral blood. However, the detailed response of the subpopulation of bone marrow and peripheral blood cells exposed to ionizing radiation has not been fully elucidated [51,52,53]. According to the literature, stem cells, T helper cells, cytotoxic T lymphocytes, monocytes, neutrophils, and B lymphocytes are all sensitive to radiation, while regulatory T cells, macrophages, dendritic cells, and NK cells (natural killer T) seem to be more resistant to radiation. There are no explicit data concerning basophilic and eosinophilic granulocytes. Erythrocytes and thrombocytes, but not their precursors, seem to be more radioresistant. Previous studies showed significant differences in the radiosensitivity of bone marrow and peripheral blood cells, and neoplastic cells. It was demonstrated that non-proliferative lymphocytes were more sensitive to radiation than proliferative lymphocytes (CD3/CD28-stimulated) [54].

NETTER-1, the first randomized study assessing the effectiveness and complications of PRRT, was performed with the use of Lutetium-177 only [20]. The authors reported a few CTCAE grade 3 or 4 hematological complications in the group of 111 patients treated with PRRT. Similar to the present study where acute complications are assessed, in the NETTER-1 long-term evaluation, the majority of hematological complications affected lymphocytes (9%). The remaining G3/G4 adverse effects referred to the platelets (2%), leukocytes (1%), and neutrophils (1%) [20]. Most of the reports in the literature on PRRT complications have been devoted almost exclusively to chronic complications. Therefore, we are not able to objectively compare such studies because the presented study presents only acute complications, which may change with the therapy duration.

Conversely, a recent study evaluating the long-term adverse effects of PRRT with [^90^Y]Y-DOTATOC and/or [^177^Lu]Lu-DOTATATE revealed a higher hematologic toxicity of PRRT in patients with baseline renal failure [24]. Svensson et al. demonstrated that the prolonged circulation of a radioisotope in patients with the renal dysfunction was probably the most important factor underlying the increased bone marrow toxicity [55]. Similarly, Bergsma et al. noted that renal function deterioration was a predictor of hematologic toxicity [56]. Their study showed subacute grade 3/4 hematologic toxicity in 34 (11%) out of 320 patients receiving [^177^Lu]Lu-DOTATATE. The toxicity persisted over six months or required blood transfusion in 50% of the patients [56]. The authors also claimed that a low baseline leukocyte count was the predictor for grade 3/4 hematologic toxicity [56]. These results are consistent with the results of other studies [24,26,29,46,47,48]. Bodei et al. demonstrated that the bone marrow reserve was depleted during subsequent [^90^Y]Y-DOTATOC cycles, particularly when the total absorbed dose exceeded 1.2 Gy [57]. However, no correlation was found between the dose absorbed by the bone marrow and acute hematologic toxicity. The above-mentioned study by Bergsma et al. suggested that the bone marrow toxicity limit was 2 Gy in the case of Lutetium-177 [56]. A study conducted to assess the complications of treatment with Yttrium-90, Lutetium-177, and the tandem (Lutetium and Yttrium) treatment revealed that myelodysplastic syndrome occurred in 2.35% of the participants [24]. Previous chemotherapy was associated with hematologic complications in only 30% of the patients. Baseline thrombocytopenia and prolonged PRRT treatment were additional risk factors. The authors also suggested the occurrence of an individual tendency towards hematologic complications in patients [24]. Sabet et al. demonstrated that reversible CTCAE G3/G4 hematologic complications were noted in 35% of the participants. However, the study group included 11 patients and the complications regressed spontaneously only in 1 out of 4 patients. Baseline parameters were achieved after two years, following the completion of PRRT [58]. Therefore, it seems that the issue requires research on a larger group of patients.

Hematologic complications may also be associated with the irradiation of the spleen, which is the main reservoir of blood cells. It was demonstrated that the spleen was one of the most irradiated organs due to the presence of somatostatin receptors on lymphocytes. The use of PRRT may lead to the reduction of cells in the peripheral blood. The present study also showed a significant decrease in the number of lymphocytes. Sabet et al. suggested that splenectomy decreased hematologic toxicity [26]. Conversely, Kulkarni et al. investigated groups of patients who underwent PRRT after splenectomy and with the spleen preserved. They found that PRRT-related hematologic toxicity did not result from the irradiation of the spleen [59]. The present study also revealed no protective factors related to previous splenectomy, which would influence the observed changes in complete blood count. Due to the lack of good dosimetry methods and their limited use in practice, the clinical parameters and improved parameters of peripheral blood are currently the most important criteria of individual PRRT planning.

Our study did not reveal a decreased glomerular filtration rate after radioisotope administration. However, a slight, but statistically significant, increase in GFR was noted in patients treated with [^90^Y]Y/[^177^Lu]Lu-DOTATATE, while in the group treated with [^177^Lu]Lu-DOTATATE, the parameter was slightly decreased and the mean creatinine concentrations slightly increased. Notably, baseline GFR was almost 12.5 mL/min/1.73 m^2^ lower in the group of patients treated with [^177^Lu]Lu-DOTATATE compared to the patients receiving [^90^Y]Y/[^177^Lu]Lu-DOTATATE (82.06 mL/min/1.73 m^2^ vs. 94.5 mL/min/1.73 m^2^). This is probably due to this fact that patients with a lower baseline GFR were qualified for treatment with [^177^Lu]Lu-DOTATATE. It is possible that the slight increase in the glomerular filtration rate in the tandem group resulted from the influence of the radioisotopes on the hemodynamic autoregulation of the renal glomeruli. 

Furthermore, we noted a significant decrease in the fractional excretion of potassium (FE K%) and an increased fractional excretion of urea (FE U%). Such changes directly indicate the early disorders of renal tubular function following the treatment. Moreover, the study revealed that albuminuria decreased several times, which may be due to the influence of radioisotopes on the glomerular filtration membrane, endothelium, and podocytes in particular. Radioisotopes may change the electric charge of the filtration membrane and make it more leakproof. This is an interesting observation and, obviously, it requires further research. The marked majority of studies conducted so far have been based on the repeated measurements of creatinine concentrations as the only parameter of renal function, thereby facilitating the calculation of the estimated glomerular filtration rate (eGFR). The assessment of the renal tubules has not been conducted until now. Therefore, the present findings may be a highly valuable parameter to assess the early phase of renal function disruption during PRRT and in the estimation of chronic kidney disease development.

Radioisotope treatment is associated with the particular exposure of the kidneys to radiation toxicity due to glomerular filtration, tubular reabsorption, and the concentration of radionuclides in the interstitium [60,61,62,63,64]. About 3% of the total radioisotope activity is reabsorbed by the proximal tubules after glomerular filtration, which leads to the long-lasting exposure of the kidney to radiation [65]. Grade 4/5 nephrotoxicity, in terms of adverse effects (end-stage kidney disease or death), was described in a maximum of 14% of the patients treated with radionuclides [66]. External radiotherapy is homogeneously administered in high doses, while systemic radioisotopes are heterogeneously distributed in body organs, so the radiation dose is markedly lower, variable in time, and characterized by an exponential decrease. With regards to [^90^Y]Y-DOTATOC and [^177^Lu]Lu-DOTATATE, doses below 40 Gy were safer for the patients with no risk factors, while in patients with the risk factors of CKD (mainly hypertension and diabetes), the dose limit of 28 Gy was recommended [40]. The supply of positively charged amino acids decreases the range of absorbed doses by 9 to 53% [34]. Amino acid solutions need to contain high quantities of lysine and arginine (up to 24 g of each) in order to provide suitable kidney protection. In the NETTER-1 study, commercially available amino acid solution, Aminosyn II 10%, was used, which contained 21.0 g of lysine, 20.4 g of arginine, and other amino acids [20]. Moreover, no cases of toxic kidney injury were observed within the median observation period of 14 months [20]. One patient was reported to have mild proteinuria and kidney function disorders, but no hospitalization or specific treatment were necessary [20]. 

In many cases, renal injury may be observed after at least six months of follow-up. Imhof et al. reported that post-PRRT glomerular filtration was decreased by about 1.8 ± 19% annually [28]. A retrospective study conducted in 1109 patients treated with Yttrium-90 revealed CTCAE (v. 4.0) grades 4 and 5 chronic kidney injury in 103 patients (9.2%) [28]. The study demonstrated that the baseline glomerular filtration rate (GFR) was the predictive factor for nephrotoxicity [28]. The study by Bodei et al., which we mentioned above, was conducted to assess the long-term PRRT tolerance in patients treated with [^90^Y]Y-DOTATOC, [^177^Lu]Lu-DOTATATE, or receiving the tandem treatment with [^90^Y]Y-DOTATATE + [^177^Lu]Lu-DOTATATE. It revealed the deterioration of renal function parameters in 34.6%, while severe nephrotoxicity occurred in 1.5% of the participants (in 30 months follow-up) [24]. Moreover, it was demonstrated that Yttrium-90 or tandem treatment were characterized by higher toxicity than treatment with Lutetium-177 only. Anemia and arterial hypertension were the most important risk factors of renal injury in this study. The authors also suggested that individual tendencies towards a toxic response to PRRT might be more due to genetic predisposition than concomitant diseases [24]. 

The present study also revealed reduced IL-18 concentrations in the urine after radioisotope administration (*p* < 0.001). No correlations were found between those changes and any of the studied factors. IL-18 is a proinflammatory cytokine. Its production increases in the kidneys in such instances as post-reperfusion renal injury [67]. The results of the available research showed its use as a biomarker of acute kidney injury in children and adolescents following cardio-surgery [68,69]. The attempts at undertaking treatment with anti-IL-18 antibodies make IL-18 a potential biomarker in monitoring such treatment. A study by Xiao et al. showed that increased IL-18 urine concentrations in irradiated primates (Macaca mulatta) depended on the dose of radiation during external radiotherapy. Low doses of ionizing radiation (5.0 Gy) during total body irradiation did not increase urine IL-18 concentrations [70]. Conversely, the application of a high dose of radiation increased urine IL-18 concentrations from day one to five, with a peak on day three following the administration of the doses of 6.5 Gy and 8.5 Gy [70]. The lack of increased urine IL-18 concentrations, or even their decrease in the present study, may be due to the immunosuppressive properties of the radioisotope within the kidneys and the reduction of the synthesis of proinflammatory molecules.

In the presented study, no hepatotoxicity of the radioisotope treatment was found. Hepatotoxicity is a rare complication of PRRT. It has been sometimes reported in patients with large and extensive liver metastases [71,72]. For example, in the NETTER-1 study, the elevation of ALT activity of grade 3 or 4 according to CTCAE was found in 3.6% and the bilirubin concentration in 1.8% of the patients [20].

Research by Bodei and Cremonesi confirmed that the use of divided doses in PRRT might reduce the risk of renal and hematologic complications [40,73]. Those studies concerned Yttrium-90. Studies conducted in patients treated with Lutetium-177 confirmed the safety of activity up to 7.4 GBq/course with no serious renal injury or hematologic complications [43]. Lutetium-177 was better tolerated by the patients in that study. The standard of NEN treatment in the world is treatment with Lutetium-177 alone. Tandem therapy is still used in clinical trials. In such situations, a combination of Lutetium-177 and Yttrium-90 with the activities of 1.85 Gbq (Lu) and 1.85 GBq (Y) can be used. In the literature, the studies confirming the effectiveness and safety of such treatment are also available [23,24,74,75].

### 4.1. Strengths of the Study

Retrospective studies constitute the majority of the available literature. The present paper is a prospective one. Studies conducted so far have largely been based on the measurements of serum creatinine as the only parameter for renal function assessment. The assessment of the renal tubules has not been conducted until now. Therefore, the present findings may be highly valuable in the assessment of the early phase of renal function disruption during PRRT and in the estimation of chronic kidney disease development.

### 4.2. Limitations of the Study

The study was conducted in a relatively low number of patients, mostly because of the low incidence of neuroendocrine neoplasms, despite the fact that the recruitment process for this study took almost two years, and the study was carried out in a center with the highest number of PRRT in Poland—and this part of Europe. The ideal situation would be one in which the results presented here should be related to the treatment effectiveness in this group of patients. However, such assessments due to long term follow-up are still in progress. Some of the epidemiological data in this group, such as the prevalence of functional NEN, pancreatic NEN, or the location of the primary lesion, may differ from the epidemiological data of a large NEN population. The carcinoid syndrome, according to different studies, occurs in 10% to 40% of NEN patients (mostly with the primary lesion in the small intestine). Pancreatic NEN constitutes approximately 30% of all GEP NENs and 60 to 90% of them are non-functioning tumors. However, our study was not an epidemiological one. It was aimed at assessing the complications of radioisotope treatment in a selected group of patients, hence probably these differences. Many of the presented results did not reach statistical significance, but only remained at the trend level, which certainly requires further research on a larger group of patients.

## 5. Conclusions

PRRT caused acute hematologic complications, particularly in the white cell line, in grade 1 and 2 according to CTCAE criteria.Tandem treatment in such assessment revealed more severe hematologic complications. In the early evaluation, PRRT did not affect glomerular filtration.However, PRRT triggered acute renal tubule dysfunction, regardless of the treatment type.In the evaluation of the acute treatment complications, PRRT appeared to be a safe option. 

## Figures and Tables

**Figure 1 jcm-11-00919-f001:**
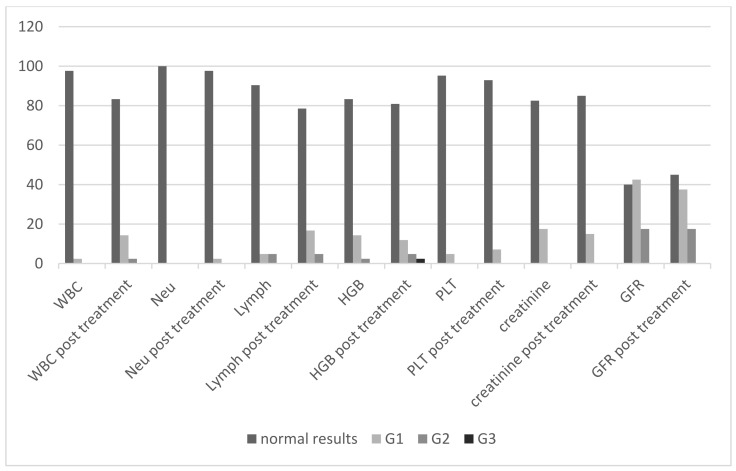
The percentage of patients before and after the radioisotope administration depending on the group of adverse events according to the CTCAE criteria. WBC—leukocytes, Neu—neutrophils, HGB—hemoglobin, PLT—platelets, GFR—glomerular filtration rate, G1, G2, G3—grades of adverse events according to the NCI CTCAE criteria.

**Table 1 jcm-11-00919-t001:** Stages of the study.

Day	Procedure
Day 1	medical history and physical examination laboratory blood and urine tests
Day 2	administration of amino acids and Ringer’s solution administration of radioisotopes
Day 3	administration of amino acids and Ringer’s solution
Day 4	laboratory blood and urine tests post-treatment scintigraphy discharge from the hospital

**Table 2 jcm-11-00919-t002:** Formulae used to calculate the indices of the fractional excretion of sodium, potassium, calcium, phosphorus, uric acid, and urea.

Fractional Excretion	Formula
FE Na%	(U_Na_ × S_cr_)/(S_Na_ × U_cr_) × 100%
FE K%	(U_K_ × S_cr_)/(S_K_ × U_cr_) × 100%
FE Ca%	(U_Ca_ × S_cr_)/(S_Ca_ × U_cr_) × 100%
FE PO_4_%	(U_PO4_ × S_cr_)/(S_PO4_ × U_cr_) × 100%
FE UA%	(U_UA_ × S_cr_)/(S_UA_ × U_cr_) × 100%
FE U%	(U_U_ × S_cr_)/(S_U_ × U_cr_) × 100%

FE—fractional excretion, U—urine concentration, S—serum concentration, cr—creatinine.

**Table 3 jcm-11-00919-t003:** Demographic characteristics of the study group.

Characteristics	Total Group (*n* = 42)	[^177^Lu]Lu-DOTATATE (*n* = 31)	[^90^Y]Y/[^177^Lu]Lu-DOTATATE (*n* = 11)
Age (years)			
mean	58.1 ± 13.1	57.8 ± 14	59 ± 11
range	23–78	23–78	43–76
Sex			
Women	19 (45.2%)	14 (45.2%)	5 (45.5%)
Men	23 (54.8%)	17 (54.8%)	6 (54.5%)
BMI (kg/m^2^)			
mean	24.9 ± 5.2	23.1 ± 5.6	24.3 ± 4.4
range	16.4–41.3	16.4–41.3	17.8–30.8
<18.5	3 (7.1%)	2 (6.5%)	1 (9.1%)
18.5–24.9	21 (50%)	14 (45.1%)	7 (63.6%)
25.0–29.9	12 (28.6%)	11 (35.5)	1 (9.1%)
≥30.0	6 (14.3%)	4 (12.9%)	2 (18.2%)
Concomitant conditions			
Chronic kidney disease	6 (14.3%)	6 (19.4%)	0 (0%)
Arterial hypertension	18 (42.9%)	13 (41.9%)	5 (45.4%)
Diabetes	12 (28.6%)	5 (16.1%)	7 (63.6%)
Hypercholesterolemia	6 (14.3%)	4 (12.9%)	2 (18.2%)
Location of the primary origin of NEN			
Pancreas	15 (35.6%)	10 (32.2%)	5 (45.4%)
small intestine	13 (30.9%)	12 (38.7%)	1 (9.1%)
large intestine	5 (12%)	3 (9.7%)	2 (18.2%)
other	5 (12%)	3 (9.7%)	2 (18.2%)
	(2 × ovary,	(2 × ovary,	(1 × stomach,
	1 × stomach,	1 × paraganglioma)	1 × lung)
	1 × paraganglioma,		
	1 × lung)		
unknown	4 (9.5%)	3 (9.7%)	1 (9.1%)
Functional NEN			
total	23 (54.8%)	15 (48.4%)	8 (72.7%)
Pancreas	10 SC (66.7%)	5 SC * (50%)	5 SC ** (100%)
small intestine	7 SC (53.8%)	6 SC (50%)	1 SC (100%)
large intestine	2 SC (40%)	1 SC (33.3%)	1 SC (50%)
ovary	1 SC (50%)	1 SC (50%)	0 SC (0%)
stomach	0 SC (0%)	0 SC (0%)	0 SC (0%)
paraganglioma	1 SC (100%)	1 SC (100%)	0 SC (0%)
lung	1 SC (100%)	0 SC (0%)	1 SC *** (100%)
unknown	1 SC (25%)	1 SC (33.3%)	0 SC (0%)
NEN histological malignancy grade [13]			
G1	20 (48%)	15 (48.4%)	5 (45.5%)
G2	22 (52%)	16 (51.6%)	6 (54.5%)
G3	0	0	0

NEN—neuroendocrine neoplasm, SC—carcinoid syndrome, PTHrP—PTH-related protein, GHRH—growth hormone-releasing hormone, *—including 1 NEN secreting PTHrP, **—including 1 NEN secreting glucagon, ***—including 1 NEN secreting GHRH.

**Table 4 jcm-11-00919-t004:** Complete blood count parameters before and after radioisotope administration (statistically significant results are in bold).

	Before Radioisotope Administration (*n* = 42)		After Radioisotope Administration (*n* = 42)				
Parameter	M	SD	M	SD	Δ	%	*p*
WBC [10^3^/µL]	6.88	1.83	5.99	1.92	−0.89	−12.936	**<0.001**
Neu [10^3^/µL]]	4.45	1.59	3.62	1.28	−0.83	−18.6517	**<0.001**
Lymph [10^3^/µL]	1.71	0.80	1.60	0.76	−0.11	−6.43275	**0.038**
RBC [10^6^/µL]	4.50	0.67	4.44	0.74	−0.06	−1.33333	0.186
HGB [g/dL]	13.40	1.87	13.21	2.02	−0.19	−1.41791	0.107
PLT [10^3^/µL]	246.19	107.04	234.50	68.57	−11.69	−4.74837	0.176
RET%	1.46	0.43	1.46	0.39	0	0	0.236

WBC—leukocytes, Neu—neutrophils, Lymph—lymphocytes, RBC—erythrocytes, HGB—hemoglobin, PLT—platelets, RET—reticulocytes, M—mean, SD—standard deviation, *p*—level of significance, Δ—change of the parameter, %—percentage of change.

**Table 5 jcm-11-00919-t005:** Complete Blood Count and biochemical parameters before and after radioisotope administration depending on the occurrence of chronic kidney disease (statistically significant results are in bold).

	GFR ≥ 60 mL/min/1.73 m^2^ (*n* = 36)					GFR < 60mL/min/1.73 m^2^ (*n* = 6)				
	Before		After			Before		After		
Parameter	M	SD	M	SD	*p*	M	SD	M	SD	*p*
WBC [10^3^/µL]	6.92	1.93	6.12	2.02	**<0.001**	6.67	1.11	5.22	0.80	**0.043**
Neu [10^3^/µL]	4.41	1.63	3.68	1.34	**<0.001**	4.66	1.48	3.26	0.83	**0.026**
Lymph [10^3^/µL]	1.78	0.81	1.66	0.78	0.182	1.32	0.62	1.26	0.57	0.329
RBC [10^6^/µL]	4.61	0.55	4.59	0.57	0.776	3.87	1.02	3.56	1.05	**0.01**
HGB [g/dL]	13.66	1.56	13.61	1.53	0.733	11.85	2.85	10.83	2.99	**0.007**
PLT [10^3^/µL]	231.39	67.29	228.47	60.27	0.739	335.00	226.31	270.67	106.31	0.27
RET%	1.42	0.39	1.44	0.39	0.539	1.71	0.58	1.58	0.41	0.205
Serum creatinine [mg/dL]	0.87	0.26	0.88	0.29	0.638	1.23	0.20	1.20	0.15	0.465
GFR _CKD-EPI cr_ [mL/min/1.73 m^2^]	91.18	21.82	90.59	23.80	0.775	51.50	11.98	54.33	13.31	0.301
FE Na%	0.69	0.69	0.85	0.54	0.26	1.89	2.54	1.44	0.54	0.663
FE K%	10.35	7.53	6.05	2.80	**0.003**	11.20	9.95	9.17	5.98	0.591
FE Ca%	0.54	0.54	2.04	6.42	0.186	0.82	0.73	0.98	0.71	0.697
FE PO_4_%	11.23	6.84	15.18	7.44	**0.005**	19.32	9.93	14.06	6.06	0.09
FE U%	34.59	20.13	52.35	14.25	**<0.001**	38.39	25.16	41.38	24.72	0.702
FE UA%	6.65	4.88	7.49	3.66	0.329	7.32	6.77	6.78	5.65	0.58
ACR [mg/g]	0.09	0.16	0.08	0.33	0.912	0.06	0.07	0.07	0.12	0.908
Albumin in the urine [mg/mL]	9.38	24.48	0.67	0.68	**0.042**	10.00	18.22	3.68	6.49	0.452
KIM-1 in the urine [pg/dL]	1713.50	1395.59	1250.04	1018.99	0.068	2966.93	1535.15	2384.20	2959.90	0.548
Il-18 in the urine [pg/mL]	150.45	118.61	44.96	52.96	**<0.001**	217.93	151.94	56.80	47.09	0.053
KIM-1 in the serum [pg/dL]	66.61	291.58	55.95	196.34	0.551	117.97	246.39	119.40	211.46	0.932
Serum albumin [mg/dL]	4.66	0.33	4.41	0.39	**<0.001**	3.92	0.88	3.55	0.77	**0.018**
ALT [IU/L]	26.44	18.79	22.38	15.16	**0.005**	22.83	14.68	21.50	14.84	0.484
Bilirubin [mg/dL]	0.67	0.44	0.79	0.49	**0.002**	0.57	0.22	0.57	0.25	1

WBC—leukocytes, Neu—neutrophils, Lymph—lymphocytes, RBC—erythrocytes, HGB—hemoglobin, PLT—platelets, RET—reticulocytes, GFR—glomerular filtration rate, cr—creatinine, FE—fractional excretion, ACR—albumin-to-creatinine ratio, KIM-1—Kidney Injury Molecule-1, IL-18—Interleukin-18, ALT—alanine aminotransferase, M—mean, SD—standard deviation, *p*—level of significance.

**Table 6 jcm-11-00919-t006:** Renal parameters before and after radioisotope administration (statistically significant results are in bold).

	Before Radioisotope Administration (*n* = 42)		After Radioisotope Administration (*n* = 42)		
Parameter	M	SD	M	SD	*p*
Serum creatinine [mg/dL]	0.93	0.28	0.93	0.29	0.317
GFR _CKD-EPI cr_ [mL/min/1.73 m^2^]	85.23	25.05	85.15	25.96	0.290
FE Na%	0.88	1.20	0.94	0.58	0.291
FE K%	10.48	7.80	6.53	3.55	**0.044**
FE Ca%	0.58	0.57	1.87	5.91	0.320
FE PO_4_%	12.50	7.84	15.00	7.18	0.098
FE U%	35.17	20.65	50.66	16.36	**<0.001**
FE UA%	5.61	0.84	5.66	0.60	0.395
ACR [mg/g]	0.08	0.15	0.08	0.31	0.527
Albumin in the urine [mg/mL]	9.47	23.45	1.12	2.64	**0.004**
KIM-1 in the urine [pg/dL]	1906.33	1469.74	1424.52	1482.91	0.147
IL-18 in the urine [pg/dL]	160.84	124.47	46.78	51.70	**<0.001**
KIM-1 in the serum [pg/mL]	74.32	282.96	65.47	197.17	0.987

GFR—glomerular filtration rate, cr—creatinine, FE—fractional excretion, ACR—albumin-to-creatinine ratio, KIM-1—Kidney Injury Molecule-1, IL-18—Interleukin-18, M—mean, SD—standard deviation, *p*—level of significance.

**Table 7 jcm-11-00919-t007:** Liver parameters before and after radioisotope administration (statistically significant results are in bold).

	Before Radioisotope Administration (*n* = 42)		After Radioisotope Administration (*n* = 42)		
Parameter	M	SD	M	SD	*p*
Serum albumin [mg/dL]	4.54	0.52	4.28	0.55	**<0.001**
ALT [IU/L]	25.90	18.12	22.25	14.93	**0.002**
Bilirubin [mg/dL]	0.66	0.41	0.76	0.47	**0.003**

ALT—alanine aminotransferase, M—mean, SD—standard deviation, *p*—level of significance.

**Table 8 jcm-11-00919-t008:** Changes in blood count and biochemical parameters depending on treatment type (statistically significant results are in bold).

	[^177^Lu]Lu-DOTATATE (*n* = 31)		[^90^Y]Y/[^177^Lu]Lu-DOTATATE (*n* = 11)		
Parameter	Δ	SD	Δ	SD	*p*
WBC [10^3^/µL]	−0.65	1.17	−1.56	1.26	**0.035**
Neu [10^3^/µL]	−0.73	1.06	−1.09	0.91	0.324
Lymph [10^3^/µL]	−0.02	0.41	−0.38	0.65	0.107
GFR _CKD-EPI cr_ [mL/min/1.73 m^2^]	−2.62	10.64	6.64	10.38	**0.018**
FE K%	−5.08	8.40	−0.69	3.76	0.121
FE U%	13.28	26.39	21.90	13.47	0.332
Albumin in the urine [mg/mL]	−4.25	17.19	−20.65	33.71	0.169
IL-18 in the urine [pg/dL]	−104.06	131.54	−130.47	79.81	0.555

WBC—leukocytes, Neu—neutrophils, Lymph—lymphocytes, GFR—glomerular filtration rate, cr—creatinine, FE—fractional excretion, IL-18—Interleukin-18, Δ—change, SD—standard deviation, *p*—level of significance.

**Table 9 jcm-11-00919-t009:** The number of patients with disrupted hematological and nephrological paremeters pre- and post-treatment (patients are categorized into grades G1–G5 according to the NCI CTCAE criteria).

	Pre-Treatment		Post-Treatment				
	G1	G2	G1	G2	G3	Total pre-treatment (%)	Total post-treatment (%)
Leukopenia	1	0	5	1	0	1/42 (2.4)	6/42 (14.3)
Neutropenia	0	0	1	0	0	0/42 (0)	1/42 (2.4)
Lymphopenia	2	2	7	2	0	4/42 (9.5)	9/42 (21.4)
Anemia	6	1	5	2	1	7/42 (16.7)	8/42 (19)
Thrombocytopenia	2	0	3	0	0	2/42 (4.8)	3/42 (7.14)
Creatinine increase	6	0	5	0	0	6/40 (15)	5/40 (12.5)
GFR decrease	17	7	15	7	0	24/40 (60)	22/40 (55)

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
