# Peer review of "Early Complications of Radioisotope Therapy with Lutetium-177 and Yttrium-90 in Patients with Neuroendocrine Neoplasms—A Preliminary Study"

_jcm, 2022, doi:10.3390/jcm11040919_

Round 1

Reviewer 1 Report

The manuscript describes a well documented study of early PRRT side effects. Hematological and kidney toxicity are well described according to multiple criteria. It appears that the combination of Yttrium-90 and Lutetium-177-labeled DOTA-TATE (1.85 GBq each) is more toxic than 177-labeled DOTA-TATE alone at a 7.4 GBq dose.

Yttrium-90 emits electrons with higher energy than lutetium-177 that have larger tracks. It was thus considered that yttrium-90 could be more appropriate for the treatment of larger tumor lesions and accordingly patients submitted to the combination were those with larger lesions. However here comes the first big weakness of the paper. How where these activities chosen? In the context, absorbed doses must be calculated to give sense to the comparison. They must also be considered for comparison with other studies.

The second big weakness comes from the fact that lutetium-177-labeled DOTA-TATE is now marketed in the USA and European under the name Lutathera. It is not acceptable that the manuscript does not mention this fact and do not cite the publication of the results of the pivotal NETTER study.

Author Response

Dear Reviewer,

Thank you very much for the nice review and extremely valuable comments. We have included them in the revised version of the manuscript. Here, there are the answers to your questions:

  1. How where these activities chosen? In the context, absorbed doses must be calculated to give sense to the comparison. They must also be considered for comparison with other studies.

 Our work was not aimed at assessing the treatment effectiveness, which would be a key issue in the context of the activity of the radioisotopes used. The aim of our study was to evaluate acute complications (not yet widely assessed in the literature) during standard radioisotope treatment of NENs. The standard PRRT regimen in this case (recommended by most scientific societies and used by most, if not all, NEN treatment centers) is to use Lutetium-177 with an activity of 7.4 GBq in 4 courses every 8 weeks. In our Center (one of the ENETS Center of Exellence), we use Lutetium alone if we do not find any large lesions (> 50 mm). Few centers, including ours, use also other type of NEN therapy - a combination of Lutetium-177 and Yttrium-90, as the so-called Tandem therapy - at our center - with activities of 1.85GBq (Lu) and 1.85GBq (Y). We use such therapy, within clinical trials, if we find at least one large lesion > 50mm. Our approach is based on many years of experience. We have been using PRRT in our center for almost 20 years (one of the first centers in the world) and we are currently the Center that carries out most such therapies in Poland and the entire Central and Eastern Europe (approx. 250 treatments per year).

Hence, the aim of the presented study was only to assess acute complications in the two most frequently used treatment regimens in our Center - using Lutetium alone or a combination of Lutetium and Yttrium, with activities that we use as standard, and resulting from our many years of experience.

  1. The second big weakness comes from the fact that lutetium-177-labeled DOTA-TATE is now marketed in the USA and European under the name Lutathera. It is not acceptable that the manuscript does not mention this fact and do not cite the publication of the results of the pivotal NETTER study.

              The lack of reference to the NETTER-1 study is an obvious mistake that escaped when editing the manuscript, and have included it in the revised version of the manuscript.

All your valuable comments and suggestions have been included in the revised version of the work.

Yours faithfully,

The Authors.

Reviewer 2 Report

This is a thorough study on the evaluation of early complications of PRRT in NEN. Although most of the information provided is already known, the research was performed prospectively proving it correct.

Suggestions and questions to address:

  1. The presentation of the paper is in general long and in some ways difficult to follow.
  2. The introduction is too long. Please keep only the relevant to the article facts.
  3. Why did the patients stay in the hospital for 4 days? Was it for the present study or is it the teams routine protocol? Usually the treatments are currently done on an outpatient basis.
  4. Why amino-acids were administered also the day after the treatment. How does this may affect the electrolytes (ie they are known to increase potassium levels)  

Author Response

Dear Reviewer,

Thank you very much for the nice review and extremely valuable comments. We have included them in the revised version of the work. Here, we provide the answer to your questions and suggestions:

  1. The presentation of the paper is in general long and in some ways difficult to follow.

According to your suggestion, we have shortened the manuscript, especially in those places that may lose a bit the narrative and the fluency of the text.

  1. The introduction is too long. Please keep only the relevant to the article facts.

By agreeing with your suggestion, we have also shortened the Introduction, leaving in it the most important issues related to the title and the aims of the work.

  1. Why did the patients stay in the hospital for 4 days? Was it for the present study or is it the teams routine protocol? Usually the treatments are currently done on an outpatient basis.

A 4-day stay at our Clinical Center is a standard for the NEN treatment. It results from a longer (2-day) administration of nephroprotective aminoacids than in other centers, as well as from our routine post-therapeutic (after 48 hours) somatostatin receptor imaging (scintigraphy), as a test confirming the purposefulness and effectiveness of the treatment. Here, we can mention that our approach is based on many years of NEN treatment experience. We have been using PRRT in our center for almost 20 years (one of the first NEN treatment centers in the world) and we are currently the center that carries out most such therapies in Poland and the entire Central and Eastern Europe (approx. 250 treatments per year), now as the one of the ENETS Center of Exellence.

  1. Why amino-acids were administered also the day after the treatment. How does this may affect the electrolytes (ie they are known to increase potassium levels)

As we mentioned above, this is due to our many years of experience. There are protocols that include a 1-day, 2-day, or even 3-day regimen of nephroprotective use of amino acids. According to our own results, and experience, all these amino-acid regimens are comparable taking into account the effects on clinical symptoms (nausea, vomiting), electrolyte concentrations, including potassium, plasma osmolality, and the acid-base balance.

Yours faithfully,

The Authors.

Round 2

Reviewer 1 Report

The manuscript reports a well-documented study of early PRRT side effects, including hematological and kidney toxicity described according to multiple criteria. It has been amended mostly to include the results of the NETTER study, which was overlooked in the first submission. The authors come to a series of conclusion. PRRT was found to cause acute hematologic complications. PRRT was not shown to affect glomerular filtration, but it PRRT triggered acute disruption of renal tubule function. These results are in line with other studies. The extent to which this study agrees or disagrees with earlier finding should be emphasized.

Despite "years of experience", it appears that a comparison between lutetium-177 and tandem treatment assessing both side effects and effectiveness is still lacking to demonstrate the interest of the tandem treatment for larger tumors. The authors state that tandem treatment showed more severe hematologic complications. This conclusion is based on a significant difference on WBC counts with a relatively high p value (0.035) and small number of patients (31 versus 11). One would accept some increase in toxicity for the benefit of better tumor control.

To the question of the selection of activities used for tandem therapy, the authors provided answers that do not make sense. The fact that this work assessed complications and not effectiveness is not relevant. Both are necessarily activity-dependent and should be related to absorbed doses. If the standard PRRT regimen for Lu-177-labeled DOTA-TATE is indeed 7.4 GBq, the combination treatment with lutetium-177 and yttrium-90 may be standard in Poland, but it is not standard worldwide. Years of experience do not substitute to a scientific approach and hopefully the 1.85 GBq lutetium-177 / 1.85 GBq yttrium-90 activities have not been haphazardly selected. The rationale of the selection, if any, must be briefly explained.

The number of patients involved in this study is relatively small (42) and this is probably why patients that received lutetium-177 only (31) and those who received the combination of radionuclides (11) are considered in most cases as a whole to describe patient demographics and occurrence of complications. Again, this is only justified if, somehow, both treatments are expected to result in similar outcomes. Only in Table 10, patients are stratified on treatment type.

Table 8 stratifies patients according to GFR status. The numbers of patients in both groups can presumably be extracted from patient data, but it would be easier to find these numbers in the text or in the Table.

Author Response

Dear Reviewer,

Thank you very much for another very thorough review. We have included your comments and suggestions in the revised version of the paper. We hope that the revised version will meet all your expectations and raise the level of the manuscript to one that is acceptable for you and publication in the JCM.

  1. We extended the discussion with your suggestions concerning a deeper comparison of our results with the data from the literature. It was not easy, because the data on acute complications of PRRT in the literature is scarce. We would like to emphasize at this point that most of the reports in the literature on PRRT complications have been devoted almost exclusively to chronic complications. Studies on acute complications are still missing. We focused on such acute complications, probably for the first time in the literature, using very sensitive markers of kidney injury (KIM-1, Il-18) as well as fractional excretion of various ions. We have thus shown (i.a. in functional tests) the effect of PRRT not only on GFR but also on very rarely, if ever, demonstrated renal tubular function.
  2. We would also like to emphasize, as we responded to your first review, that the standard of NEN treatment in our Center, as well as around the world, is treatment with Lutetium-177 alone. Tandem therapy is only used in clinical trials. Many years of experience have allowed only, after various tests (schemes and radioisotope activities), to select for clinical trials activities that, in our opinion, should be further assessed. We have presented the method of selecting patients for both types of treatment. However, as suggested by you, we extended the description of patient qualifications to both PRRT arms with more detailed data.
  3. You are absolutely right that the ideal situation would be the one in which the results presented here should be related to the effectiveness of treatment in this group of patients. Such research is still ahead of us, some are in progress, others are planned. The aim of this study was to assess acute complications, rarely shown in the literature so far. We have presented a fragment of our work to date, presenting information about the complications found in this group of patients.
  4. In line with your suggestions, we also supplemented all tables with the size of the groups to make them and the presentation of the results more readable.

Yours faithfully,

Kornelia Zareba and Co-authors

Reviewer 2 Report

Thank you for your appropriate responses

Author Response

Dear Reviewer,

We want to thank you for all valuable remarks. We are happy that you enjoyed this topic.

Yours faithfully,

Kornelia Zareba and Co-authors